# Clinical usability of 3D gradient-echo-based ultrashort echo time imaging: Is it enough to facilitate diagnostic decision in real-world practice?

**So Hyeon Bak[1,2☉], Jinil Park[3☉], Seokwon Lee[4], Jong Hee Kim[1], Ho Yun Lee[1,5]\*, Jang-Yeon Park[4,3]\***

**1** Department of Radiology and Center for Imaging Science, Samsung Medical Center, Sungkyunkwan University School of Medicine, Seoul, Republic of Korea, **2** Department of Radiology and Research Institute of Radiology, University of Ulsan College of Medicine, Asan Medical Center, Seoul, Republic of Korea, **3** Department of Intelligent Precision Healthcare Convergence, Sungkyunkwan University, Suwon, Republic of Korea, **4** Department of Biomedical Engineering, Sungkyunkwan University, Suwon, Republic of Korea, **5** Department of Health Sciences and Technology, Samsung Advanced Institute for Health Sciences & Technology (SAIHST), Sungkyunkwan University, Seoul, Korea

☉ These authors contributed equally to this work.
\* hoyunlee96@gmail.com (HYL); jyparu@skku.edu (J-YP)

**Data Availability Statement:** Data cannot be shared publicly because of patient privacy.

**Funding:** This work was supported by NRF-2020R1A2B5B02002676, NRF-2021R1A4A5032806, NRF-2021R1C1C2008365,

# Abstract

## Background

With recent advances in magnetic resonance imaging (MRI) technology, the practical role of lung MRI is expanding despite the inherent challenges of the thorax. The purpose of our study was to evaluate the current status of the concurrent dephasing and excitation (CODE) ultrashort echo-time sequence and the $T_1$-weighted volumetric interpolated breath-hold examination (VIBE) sequence in the evaluation of thoracic disease by comparing it with the gold standard computed tomography (CT).

## Methods

Twenty-four patients with lung cancer and mediastinal masses underwent both CT and MRI including $T_1$-weighted VIBE and CODE. For CODE images, data were acquired in free breathing and end-expiratory images were reconstructed using retrospective respiratory gating. All images were evaluated through qualitative and quantitative approaches regarding various anatomical structures and lesions (nodule, mediastinal mass, emphysema, reticulation, honeycombing, bronchiectasis, pleural plaque and lymphadenopathy) inside the thorax in terms of diagnostic performance in making specific decisions.

## Results

Depiction of the lung parenchyma, mediastinal and pleural lesion was not significant different among the three modalities ($p > 0.05$). Intra-tumoral and peritumoral features of lung nodules were not significant different in the CT, VIBE or CODE images ($p > 0.05$). However, VIBE and CODE had significantly lower image quality and poorer depiction of airway, great

and NRF-2022R1A2C1003999. None of the authors have potential conflicts of financial relationships to disclose. The funders had no role in study design, data collection and analysis, decision to publish, or preparation of the manuscript.

**Competing interests:** None of the authors have potential conflicts of interest to disclose.

vessels, and emphysema compared to CT ($p < 0.05$). Image quality of central airways and depiction of bronchi were significantly better in CODE than in VIBE ($p < 0.001$ and $p = 0.005$). In contrast, the depiction of the vasculature was better for VIBE than CODE images ($p = 0.003$). The signal-to-noise ratio (SNR) and contrast-to-noise ratio (CNR) were significant greater in VIBE than CODE except for $SNR_{lung}$ and $SNR_{nodule}$ ($p < 0.05$).

## Conclusions

Our study showed the potential of CODE and VIBE sequences in the evaluation of localized thoracic abnormalities including solid pulmonary nodules.

## Introduction

Concerns about radiation exposure related to computed tomography (CT) of children, pregnant women, and patients requiring repeated examination over prolonged periods have led to increased interest in magnetic resonance imaging (MRI) as alternative non-radiation imaging modality [1,2]. However, lung MRI has low signal intensity in aerated lungs due to low proton density and fast signal decay [3]. Recent advances in MRI pulse sequences, acceleration methods such as parallel imaging, post-processing software, and analytical methods have improved the utility of MRI for diagnosis of thoracic diseases [4] (Supplementary Methods).

Among the advanced pulse sequences for lung MRI, 3D ultrashort echo-time (UTE) imaging sequences are impressive as they provide highly sensitive signals across the entire lung area during free breathing and enable reliable assessment of lung tissue density by reducing signal loss in the lung parenchyma thanks to the very short echo time (TE) [5,6]. In terms of the signal acquisition scheme, free-induction-decay (FID) signals are usually acquired in the shape of a Koosh ball in most 3D UTE imaging sequences, such as original 3D UTE versions [7], zero echo-time (ZTE) sequences [8], and pointwise-encoding time reduction with radial acquisition (PETRA) sequences [9]. On the other hand, asymmetric gradient echoes are acquired in a 3D UTE sequence called concurrent dephasing and excitation (CODE) [10]. When compared with FID-based UTE sequences, the CODE sequence better defines the $k$-space center in image reconstruction by preserving the echo peak while slightly increasing TE ($\geq \sim 140\ \mu s$), and is easy to implement on clinical scanners without additional hardware requirements. According to previous studies [11,12], CODE provides a better signal-to-noise ratio (SNR; $\sim 16\%$) than the original 3D UTE sequence in phantom and similar image quality in lung images, and showed better image quality than conventional lung MRI sequences such as $T_2$-weighted turbo spin echo (TSE) [11] and $T_1$-weighted volumetric interpolated breath-hold examination (VIBE) [11,12] for well-controlled healthy subjects. CODE images are obtained in free breathing with retrospective respiratory gating, alleviating the burden on patients with lung disease, especially those who have difficulty holding their breath.

The purpose of this study was to compare CODE and $T_1$-weighted VIBE, the most widely used post-contrast MR sequence, with the gold-standard CT to determine the applicability of MR in the evaluation of thoracic disease and ultimately to confirm the feasibility of CODE in real-world practice.

## Methods

### Patient population

The Institutional Review Board of our institute approved this prospective study (approval #2018-01-099), and all patients were required to provide written informed consent. Twenty-six consecutive patients underwent chest MRI from June 2019 to January 2020. Because we excluded two patients without chest CT data, a total of 24 patients with both chest CT and MRI data were included in this study. All patients underwent a clinically ordered MRI scan to evaluate lung cancer ($n$ = 20) or mediastinal mass ($n$ = 4) (Fig 1).

### CT acquisition

All CT images were obtained with a 64-detector-row CT scanner (LightSpeed VCT; GE Healthcare, Waukesha, WI, USA) using the following parameters: detector collimation, 1.25 or 0.625 mm; field of view, 36 cm; tube current, 125 mA; tube voltage, 120 kVp; beam width, 10–20 mm; beam pitch, 1.375–1.5; section thickness, 2.5 mm; and matrix size, 512×512 mm. The scan coverage was from the supraclavicular fossa to the mid-portion of kidneys. All CT images were acquired with contrast enhancement in the supine position at full inspiration. CT

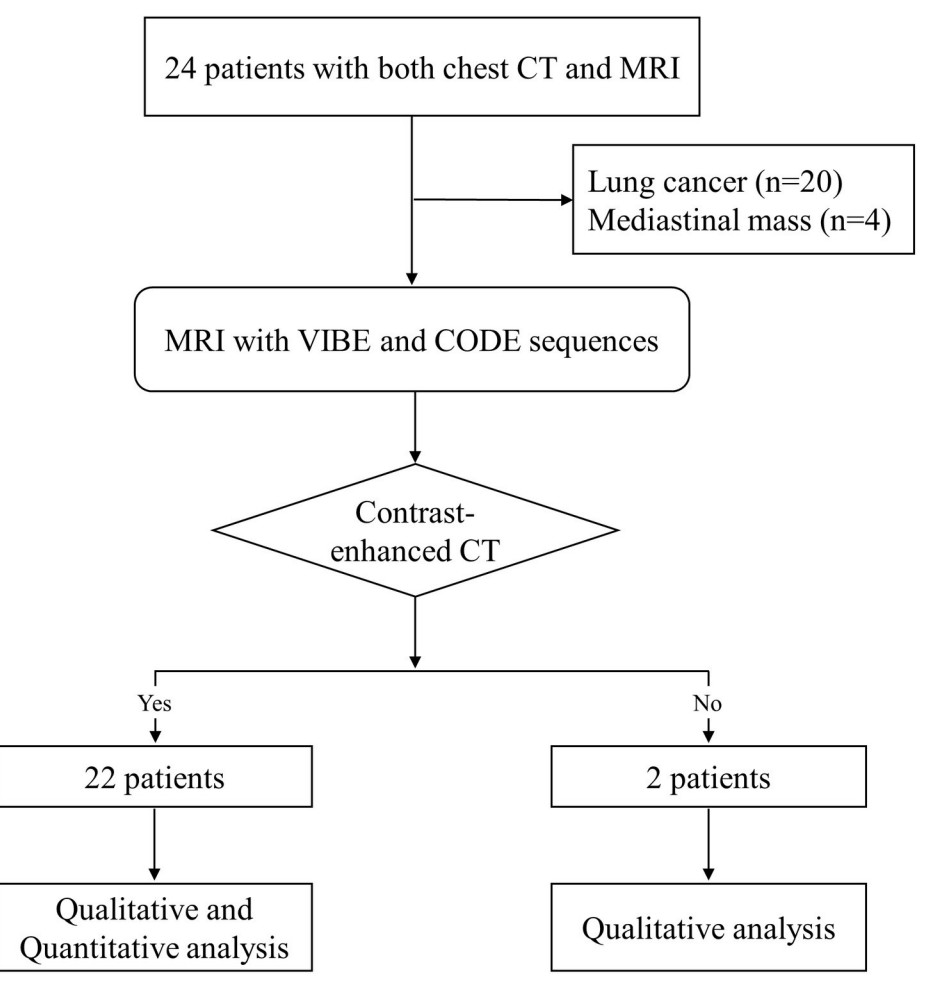

**Fig 1. Flow chart for study population selection.**

scanning was obtained 90 seconds after the administration of contrast agent at a rate of 1.5 mL/s using a power injector, which was followed by a 20 cm$^3$ saline flush at a rate of 1.5 mL/sec. The image data were reconstructed with a soft-tissue algorithm for the mediastinal window image and a bone algorithm for the lung window image. Both mediastinal (width, 400 Hounsfield units [HU]; level, 20 HU) and lung (width, 1,500 HU; level, -700 HU) window images were displayed for assessment of image quality and lesions.

## MRI acquisition

All MRI was acquired using a 3T clinical MRI system (MAGENTOM Skyra; Siemens Medical Solution, Erlangen, Germany). The scan coverage was from the supraclavicular fossa to the middle of the kidney. Fat-suppressed VIBE and CODE were performed with gadolinium-based contrast agent injection using an 18-chanel chest coil and a 16-channel spine coil.

VIBE is a fast $T_1$-weighted 3D gradient-echo imaging sequence performed during breath-hold (Fig 2A). CODE is a 3D gradient-echo-based UTE sequence in which an initial dephasing of the readout gradient is performed while applying a frequency-selective radiofrequency (RF) pulse to minimize TE in the 3D radial acquisition scheme (Fig 2B) [10]. CODE data were acquired under free breathing conditions. The acquired data included self-navigation echoes, from which respiratory motion signals were extracted for retrospective respiratory gating. VIBE and CODE sequences acquisition are detailed in the supplement. All experimental parameters in CODE and VIBE imaging are summarized in S1 Table.

## Image reconstruction

All CODE images were reconstructed offline with a home-built program using MATLAB (R2011a, MathWorks, Natick, MA, USA). A 3D fast Fourier transform (FFT) was performed

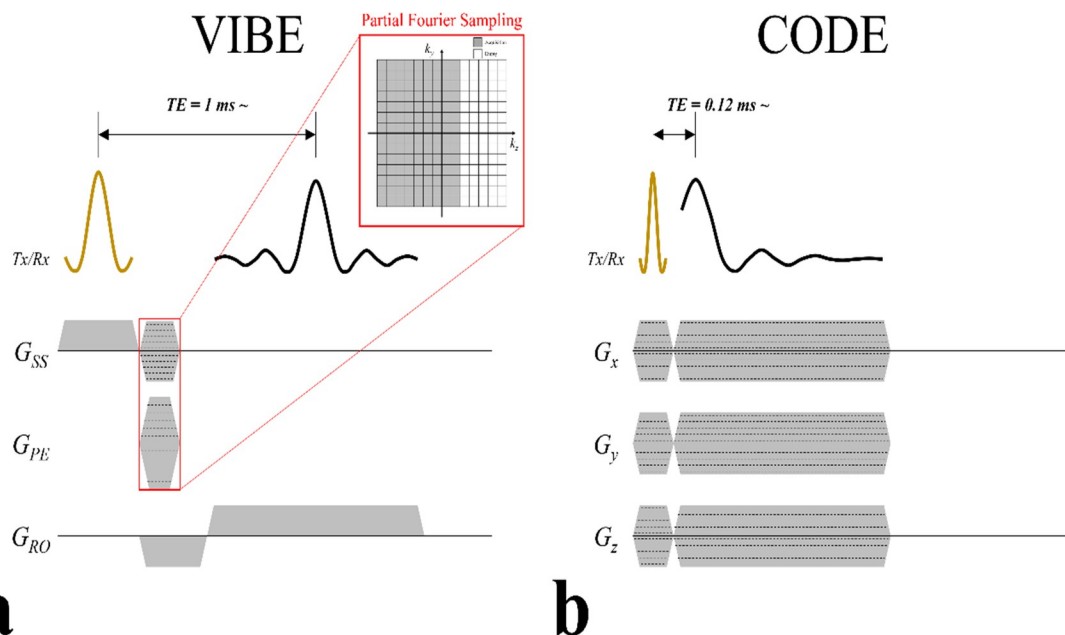

**Fig 2. Schematic diagram of the VIBE and CODE sequences.** (A) The VIBE sequence is basically same as a conventional 3D gradient-echo-imaging sequence, except for asymmetric sampling of the phase-encoding steps with zero padding in the slab-selective direction. (B) The CODE sequence is a 3D radial asymmetric gradient echo-based UTE sequence. Due to the absence of a phase-encoding step, CODE has a much shorter TE and is less sensitive to motion than VIBE.

after a gridding procedure that distributed the 3D radially sampled data in the Cartesian coordinates system by applying a Kaiser-Bessel kernel function for interpolation. The $k$-space center was defined by locating the average peak of acquired asymmetric echoes.

End-expiratory CODE images were reconstructed via retrospective respiratory gating, i.e. by selecting the radial views corresponding to the end-expiratory phase of the respiratory signal. End-expiratory images were chosen because expiration typically lasts longer than inspiration and therefore end-expiratory data has a greater number of radial views for image reconstruction. In addition, further reconstruction of the CODE images was performed with a slab thickness similar to that for VIBE images. The CODE sequence is a 3D image technique, and the resolution of the image used in this paper is iso-tropic and is $0.82 \times 0.82 \times 0.82 mm^3$. The high isotropic resolution has the advantage of being able to observe detailed anatomical structures in three planes (axial, coronal, and sagittal), but it is difficult to observe changes in blood vessels and lesions in through-plane direction. To improve this problem, we apply a sliding window averaging 4 voxels (size, $0.82 \times 0.82 \times 3.28 mm^3$) in the direction corresponding to the through-plane of the VIBE image. This type of averaging window, with tight sliding applied to every single voxel, allows for a more continuous change in the averaging direction than simple averaging of every four voxels.

## Qualitative image assessment

The evaluation of MRI was based on CT images as a reference standard. Axial CT, VIBE and CODE images were used for the analyses. In 24 patients, lesions such as nodules ($N = 23$), mediastinal masses ($N = 4$), emphysema ($N = 9$), reticulation ($N = 4$), honeycombing ($N = 2$), bronchiectasis ($N = 4$), pleural plaque ($N = 1$), and lymphadenopathy ($N = 3$) were detected on chest CT images.

A chest radiologist with thirteen years of experience assessed the image quality of the CT, VIBE and CODE. The score of each category was rated on a five-point scale [13,14] and the degree of emphysema was rated on a four-point scale (S2 Table) [15]. Attenuation of nodules was classified as solid nodules, part-solid nodules, and ground-glass nodules (GGN) [16]. The long-axis diameter of nodules was measured in an axial image, with the long axis defined perpendicular to the short axis [17]. The morphology of nodules was classified as spherical or non-spherical. The margins of nodules were classified as smooth, irregular non-smooth, spiculated, or lobulated. All nodules were evaluated for the presence of air bronchograms, internal calcification, thickening of the bronchovascular bundle, and pleural tags. [16] Thickening of the bronchovascular bundle was defined as irregular thickening of peribronchovascular interstitium adjacent to the nodule [18,19].

## Quantitative image assessment

Since two of 24 patients underwent non-contrast CT, quantitative assessment was performed on 22 patients who underwent contrast-enhanced CT. The SNR and contrast-to-noise ratio (CNR) were calculated for regions of interest (ROI) in background air, and pulmonary artery truncus, lung parenchyma, and lung nodules. Three regions in the axial plane were selected to measure the signal intensity (SI) in the lung parenchyma at the level of the aortic arch, pulmonary trunk, and inferior pulmonary vein in the left and right lungs. When measuring the SI of the lung parenchyma, vascular markings and fissures were avoided. The data from the six resulting ROIs were averaged to calculate the SI of the lung. The SI of pulmonary artery was measured in the pulmonary trunk. Noise was measured as the standard deviation (SD) of the SI at an ROI of 600 mm$^2$ outside the object. The SNR was calculated as the SI of the lungs, pulmonary trunks, and nodules divided by the noise. The CNR was calculated by subtracting the

SI of the lung parenchyma from the SI of the pulmonary artery and nodule, respectively, and dividing the value by the noise [13]. CODE measured SNR and CNR from the original image (0.812 mm).

## Statistical analyses

All continuous variables were expressed as the mean ± SD. A one-way analysis of variance (ANOVA) test was performed to assess differences between imaging modalities and sequences. For all statistical analyses, $p$ values less than 0.05 were considered statistically significant. All statistical analyses were performed using SPSS® (Version 23.0, SPSS Inc., Chicago, IL, USA).

## Results

This study included 24 patients (17 males, 7 females; mean age, 60.3 ± 15.8 years; range, 16.0– 77.0 years). All patients completed an MRI examination without significant adverse effects. The mean interval between the MRI and chest CT was 10.0 ± 13.1 days. The mean scan time was 12.38 sec for the MRI with VIBE and 10 min 40 sec for MRI with CODE.

### Qualitative image assessment

The results of the qualitative assessment of CT and MRI are summarized in Table 1. The CT imaging provided excellent image quality for normal structures and lesions (5.0±0.0), while VIBE and CODE MRI had significantly lower image quality and poorer depiction of airways, great vessels, and emphysema ($p < 0.05$). Except for emphysema, there were no significant differences in image quality of the lung parenchymal, mediastinal and pleural lesions among the

**Table 1. Qualitative assessment of normal pulmonary structures and specific lesions in images obtained using CT, or MRI with VIBE or CODE sequences.**

| | CT | VIBE | CODE | $p$ value | $p$ value (CT vs. VIBE) | $p$ value (CT vs. CODE) | $p$ value (VIBE vs. CODE) |
|---|---|---|---|---|---|---|---|
| Airway evaluation | | | | | | | |
| Image quality of large airway ($N = 24$) | 5.0 ± 0.0 | 3.0 ± 0.8 | 4.1 ± 0.4 | < 0.001 | < 0.001 | < 0.001 | < 0.001 |
| Depiction of the bronchus ($N = 24$) | 5.0 ± 0.0 | 3.4 ± 0.7 | 3.9 ± 0.5 | < 0.001 | < 0.001 | < 0.001 | 0.005 |
| Depiction of bronchiectasis ($N = 4$) | 5.0 ± 0.0 | 4.3 ± 1.0 | 4.5 ± 0.6 | 0.296 | 0.278 | 0.540 | 0.850 |
| Vessel evaluation | | | | | | | |
| Image quality of great vessels ($N = 24$) | 5.0 ± 0.0 | 4.2 ± 0.5 | 4.0 ± 0.4 | < 0.001 | < 0.001 | < 0.001 | 0.429 |
| Depiction of intrapulmonary vasculature ($N = 24$) | 5.0 ± 0.0 | 5.0 ± 0.0 | 4.8 ± 0.4 | 0.010 | 1.000 | 0.003 | 0.003 |
| Depiction of lung parenchyma lesions | | | | | | | |
| Lung nodule ($N = 23$) | 5.0 ± 0.0 | 4.9 ± 0.4 | 5.0 ± 0.0 | 0.373 | 0.443 | 1.000 | 0.443 |
| Reticulation ($N = 4$) | 5.0 ± 0.0 | 4.0 ± 1.2 | 4.0 ± 1.2 | 0.274 | 0.336 | 0.336 | 1.000 |
| Honeycombing ($N = 2$) | 5.0 ± 0.0 | 5.0 ± 0.0 | 5.0 ± 0.0 | 1.000 | 1.000 | 1.000 | 1.000 |
| Emphysema ($N = 9$) | 5.0 ± 0.0 | 3.7 ± 1.3 | 3.9 ± 1.4 | 0.036 | 0.042 | 0.101 | 0.904 |
| Emphysema score ($N = 24$) | 2.6 ± 0.5 | 1.8 ± 0.4 | 1.9 ± 0.3 | 0.002 | 0.003 | 0.010 | 0.855 |
| Depiction of mediastinal lesions | | | | | | | |
| Mediastinal mass ($N = 4$) | 5.0 ± 0.0 | 5.0 ± 0.0 | 5.0 ± 0.0 | 1.000 | 1.000 | 1.000 | 1.000 |
| Lymphadenopathy ($N = 3$) | 5.0 ± 0.0 | 4.7 ± 0.6 | 4.7 ± 0.6 | 0.630 | 0.679 | 0.679 | 1.000 |
| Depiction of pleural lesion | | | | | | | |
| Plaque ($N = 1$) | 5.0 ± 0.0 | 3.0 ± 2.8 | 3.0 ± 2.8 | 0.650 | 0.695 | 0.695 | 1.000 |

Data are presented as the mean ± standard deviation.

Abbreviations: CODE, concurrent dephasing and excitation; CT, computed tomography; VIBE, volumetric interpolated breath-hold.

three modalities ($p > 0.05$, Figs 3 and 4). When VIBE and CODE images were compared, the image quality of the central airway and depiction of bronchi were significantly better in CODE (4.1±0.4 and 3.9±0.5) than in VIBE (3.0±0.8 and 3.4±0.7, $p < 0.001$ and $p = 0.005$) images (Fig 5). In contrast, depiction of the intrapulmonary vasculature was better in VIBE (5.0±0.0) than in CODE images (4.8±0.4, $p = 0.003$). However, when reconstructed with the same slab thickness as used in VIBE imaging, the depiction of intrapulmonary vasculature in CODE images was not significantly different from that in VIBE and CT images ($p = 0.75, 0.75$, respectively).

Three of 23 nodules were subsolid nodules (two part-solid nodule, one pure ground-glass nodule). Depiction of the 23 nodules was not significantly different in the CT, VIBE, or CT images ($p = 0.373$). There were no statistically significant differences in the long- and short-axis diameters of the nodules, but both measurements were underestimated in VIBE images (CT, 2.3±0.8 and 1.7±0.6 cm; VIBE, 2.2±0.9 cm, and 1.5±0.7 cm; CODE, 2.3±0.9 cm and 1.6

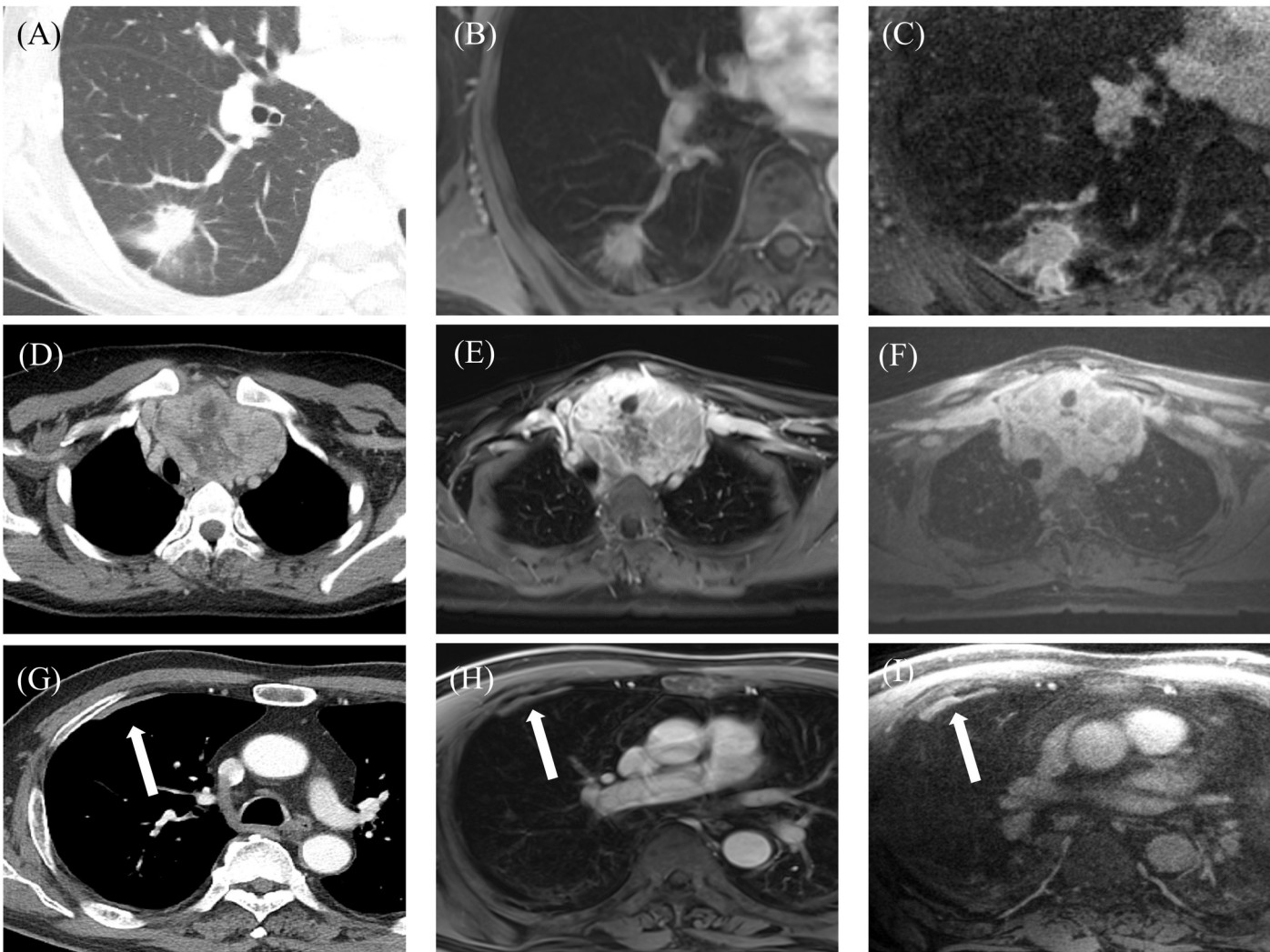

**Fig 3.** Chest images from a 63-year-old female with lung cancer in the right lower lobe showing spiculated margins, internal air and pleural tags obtained using CT (A), or MRI with VIBE (B) or CODE (C). In a 24-year-old female, CT (D), VIBE (E), and CODE (F) images each show a huge heterogeneously enhanced mass narrowing the lumen of the trachea in the anterior mediastinum. Images from a 71-year-old male indicate pleural plaque (white arrow) in the right hemithorax on CT (G), while both VIBE (H) and CODE (I) imaging also show pleural plaque (white arrow). CODE produced a stronger signal and the plaque is clearly visible.

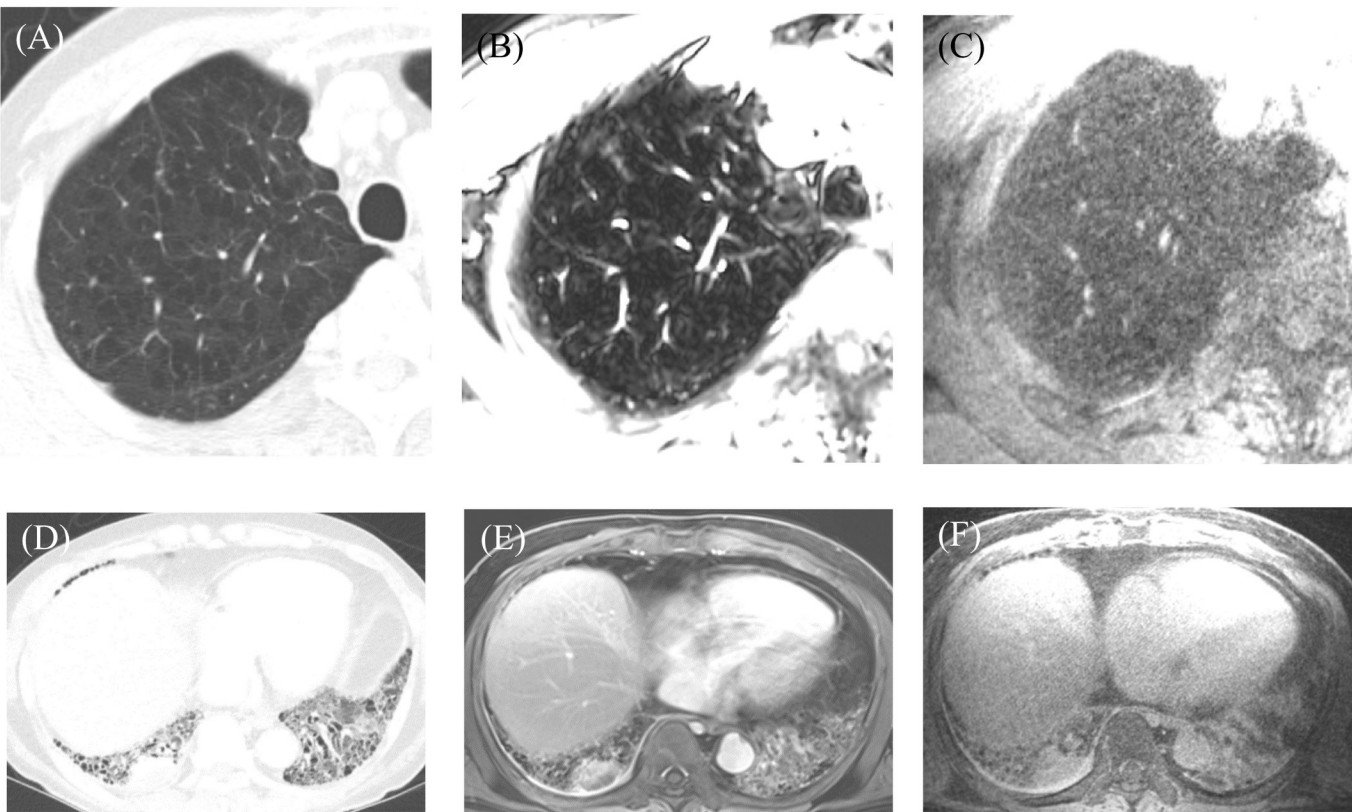

**Fig 4.** (A-C) Images from a 75-year-old male with emphysema. Chest CT (A) shows centrilobular and paraseptal emphysema. Although emphysema is visible in images obtained using MRI with VIBE (B) or CODE (C) sequences, the extent of emphysema is underestimated on CODE imaging. (D-F) Images from a 75-year-old male with idiopathic pulmonary fibrosis and lung cancer. Chest CT (D) shows a subpleural nodule in right lower lobe and honeycombing, traction bronchiectasis, and reticulation in the both lower lobe. The VIBE (E) and CODE (F) images also indicate a subpleural nodule and honeycombing in the right lower lobe, but bronchiectasis and reticulation in the left lower lobe were less clear in the VIBE and CODE images as compared with the CT image.

±0.6 cm; $p$ = 0.871 and 0.837, respectively). Other intra-tumoral and peritumoral features were not significantly different in the CT, VIBE, or CODE images ($p > 0.05$; Table 2 and Fig 4).

## Quantitative image assessment

Table 3 summarizes the SNR and CNR of the VIBE and CODE images. The SNR of the pulmonary artery ($SNR_{PA}$), the CNR of the pulmonary artery ($CNR_{PA}$), and the CNR of the lung nodules ($CNR_{nodule}$) were greater in VIBE than in CODE images ($p < 0.001$). In contrast, the SNR of the lung parenchyma ($SNR_{lung}$) in the VIBE images was significantly lower than that in the CODE images (21.1 ± 12.9, 91.2 ± 12.3, respectively, $p < 0.001$). There were no significant differences between the $SNR_{nodule}$ of the VIBE and CODE images (302.2±131.8, 233.9±115.5, respectively, $p = 0.056$).

## Discussion

While FID-based UTE sequences are commonly used to provide better visualization of endogenous lung parenchyma than conventional lung MRI sequences [4], a previous study has shown that the asymmetric echo-based CODE sequence was also sufficient to visualize lung structures in healthy subjects, despite slightly longer echo times than FID-based UTE sequences due to gradient-echo acquisition [11]. However, the feasibility of the CODE

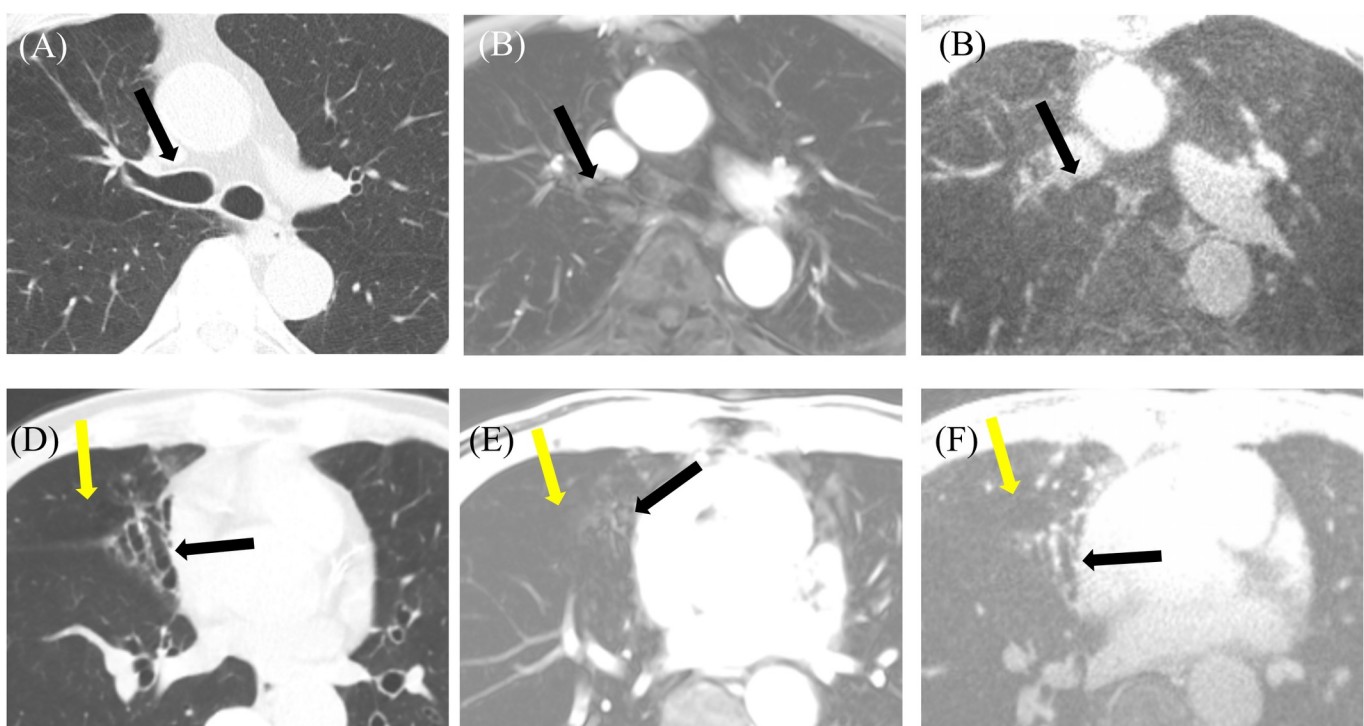

**Fig 5.** Chest CT (A), VIBE (B), and CODE (C) MRI at the level of both main bronchi (black arrow) in a 59-year-old male. Both main bronchi appear normal in the chest CT and CODE images, but it appears that there is stenosis in the right main bronchus due to an artifact in the VIBE image. (D-E) The chest CT (D) showed bronchiectasis (black arrow) in the right middle lobe and air trapping (yellow arrow) in the adjacent right upper lobe in a 73-year-old male. Bronchiectasis (black arrow) and air trapping (yellow arrow) are less clearly visible in the VIBE image (E) due to an internal artifact, but is more clearly visible in the CODE image (F).

sequence in patients with various thoracic diseases has not been investigated. Therefore, this study was intended to evaluate the current status of the CODE and VIBE as an independent diagnostic imaging tool for various thoracic disorders frequently encountered in routine clinical practice compared with CT technique.

**Table 2. Comparison of nodule characteristics in images obtained using CT, or MRI with VIBE or CODE sequences.**

|  | CT | VIBE | CODE | *p* value | *p* value (CT vs. VIBE) | *p* value (CT vs. CODE) | *p* value (VIBE vs. CODE) |
|---|---|---|---|---|---|---|---|
| Depiction of nodule | 5.0 ± 0.0 | 4.9 ± 0.4 | 5.0 ± 0.0 | 0.373 | 0.443 | 1.000 | 0.443 |
| Intratumoral features |  |  |  |  |  |  |  |
| Solidity | 1.2 ± 0.5 | 1.3 ± 0.6 | 1.0 ± 0.2 | 0.297 | 0.807 | 0.619 | 0.269 |
| Long-axis diameter | 2.3 ± 0.8 | 2.2 ± 0.9 | 2.3 ± 0.9 | 0.871 | 0.906 | 0.998 | 0.879 |
| Short-axis diameter | 1.7 ± 0.6 | 1.5 ± 0.7 | 1.6 ± 0.6 | 0.837 | 0.824 | 0.966 | 0.937 |
| Morphology | 1.8 ± 0.4 | 1.8 ± 0.4 | 1.8 ± 0.4 | 0.913 | 0.928 | 0.928 | 1.000 |
| Margin | 3.0 ± 0.9 | 2.9 ± 1.0 | 2.8 ± 1.0 | 0.672 | 0.896 | 0.646 | 0.896 |
| Air bronchogram | 1.4 ± 0.5 | 1.5 ± 0.5 | 1.4 ± 0.5 | 0.844 | 0.954 | 0.954 | 0.830 |
| Calcification | 1.1 ± 0.3 | 1.0 ± 0.2 | 1.1 ± 0.3 | 0.532 | 0.560 | 0.964 | 0.864 |
| Peritumoral features |  |  |  |  |  |  |  |
| Bronchovascular bundle thickening | 1.3 ± 0.4 | 1.4 ± 0.5 | 1.4 ± 0.5 | 0.774 | 0.809 | 0.809 | 1.000 |
| Pleural tag | 1.4 ± 0.5 | 1.5 ± 0.5 | 1.4 ± 0.5 | 0.844 | 0.954 | 0954 | 0.830 |

Data are presented as the mean ± standard deviation.

Abbreviations: CODE, concurrent dephasing and excitation; CT, computed tomography; VIBE, volumetric interpolated breath-hold.

**Table 3. Quantitative comparison of contrast-to-noise and signal-to-noise ratio in images obtained using VIBE and CODE sequences.**

| | VIBE | CODE | *p* value |
|---|---|---|---|
| $SNR_{lung}$ | 21.1 ± 12.9 | 91.2 ± 12.3 | < 0.001 |
| $SNR_{PA}$ | 592.7 ± 194.1 | 332.8 ± 76.0 | < 0.001 |
| $SNR_{nodule}$ | 302.2 ± 131.8 | 233.9± 115.5 | 0.056 |
| $CNR_{PA}$ | 571.5 ± 186.6 | 241.6 ± 70.4 | < 0.001 |
| $CNR_{nodule}$ | 280.9 ± 124.3 | 143.5 ± 112.1 | < 0.001 |

Data are presented as the mean ± standard deviation.

Abbreviations: CNR, contrast-to-noise ratio; CODE, concurrent dephasing and excitation; CT, computed tomography; SNR, signal-to-noise ratio; VIBE, volumetric interpolated breath-hold examination.

A previous study reported that PETRA, one of the FID-based UTE sequences, provided image quality similar to CT in airway disease assessment, allowing consistent bronchial viewing up to the level of the fourth-generation airways [20]. In our study, CODE was able to evaluate down to the subsegmental bronchial level, whereas VIBE showed artifacts in the main bronchi in several cases, which limited evaluation at the segmental bronchial level. In addition, the delineation of bronchiectasis on the CODE image was superior to that of the VIBE image. Similar to our study, another previous study reported that the UTE sequence provided superior delineation of mild bronchiectasis and precise characterization of saccular or cylindrical bronchiectasis [21]. The finding that the CODE sequence provides better airway evaluation than the VIBE sequence may be mainly attributed to the much shorter echo time; in other words, because the field inhomogeneity of the airway is greater than that of other tissues due to the large difference in magnetic susceptibility between the airway wall and air, the sub-millisecond echo time of CODE is able to better delineate the airway while reducing signal loss.

In a preliminary study of healthy volunteers, [11] imaging using the CODE sequence was superior to VIBE in terms of overall image quality, including better delineation of internal lung structures. However, in this study, CODE was superior to VIBE for airway evaluation, but inferior for great vessel and pulmonary vasculature evaluation when assessed at its original isotropic resolution ($0.82^3$ mm$^3$). In CODE images, there was some blurring of the margins of the great vessels, which reduced the image quality slightly compared with that of CT or VIBE imaging; however, the quality of the CODE images did not limit evaluation of the great vessels. Since several studies showed that maximum intensity projection (MIP) with an appropriate slab thickness could improve the detection efficiency of pulmonary nodules on CT [22,23], the CODE technique was further evaluated with a slab thickness (= 3.28 mm) similar to VIBE (= 3 mm) in terms of pulmonary vasculature. In this case, depiction of the pulmonary vasculature in CODE images was not significantly different from that obtained using CT or VIBE.

MRI using a variety of imaging sequences can be useful to characterize the malignant potential, and may be superior to CT in cases such as T descriptor evaluation because of its excellent soft tissue contrast [2,24]. The current study shows that the depiction and morphologic evaluation of solid nodule in CODE and VIBE were not significant different from that in CT image. The depiction of part-solid nodules and purely ground-glass opacity nodules may be underestimated by MR [4,16], yet the number of part-solid nodules and purely ground-glass opacity nodules was too small to evaluate nodule detection and morphologic characterization in our study. In addition, it has been reported that UTE sequences could improve the detection rate of small nodules, especially for nodules larger than 4 mm, for which the detection rate was improved by more than 90% [2,16,25]; however, due to smooth the margins of nodule, the nodule dimensions were underestimated by about 1–2 mm [16]. In our study,

when compared with chest CT, CODE showed similar performance in the detection and morphological characterization of nodules larger than 5 mm at the nominal isotropic resolution (e.g., $0.82^3$ mm$^3$). In contrast, it seemed that VIBE underestimated the long- and short-axis diameters of the nodules by approximately 1–2 mm as shown in other studies, but this finding was not statistically significant. These results may vary depending on the slice thickness. Thicker slices may lead to greater measurement variability and smooth the margins of nodule due to partial volume effects [26]. In this study, CODE had an advantage over VIBE because the slice thickness for CODE images was 0.82 mm, which was much thinner than the 3 mm slice thickness for VIBE images.

UTE imaging sequences can provide reliable evaluation of lung tissue structures and changes in lung tissue density by maximizing signal sensitivity and reducing signal loss in the lung parenchyma [5,6,27]. Lederline et al. [5] showed that 3D UTE imaging could provide signals within the lung parenchyma and quantitatively assess lung tissue density or lung water content. In other studies, the efficacy of pulmonary UTE imaging for diagnosis of most radiological findings, except emphysema or bullae, bronchiectasis, and reticular opacity, was not significantly different from that of CT [28], and MRI scores showed similar accuracy to CT in predicting interstitial lung disease (ILD) progression [29]. In our study, CODE imaging improved the SNR of the lung parenchyma as compared with VIBE, and the diagnosis of parenchymal lesions except for emphysema did not differ significantly from CT. These results were expected because the CODE sequence has a much shorter TE than VIBE, but less $T_1$ weighting. On the other hand, since CODE images were reconstructed by gating the end-expiratory data, the extent of ILD or emphysema in CODE images was different from the results obtained by CT at full inspiration. Moreover, CODE images tend to underestimate the detection and extent evaluation of emphysema in the upper lobe due to noise, which may be due to incomplete RF excitation of the apical region. CODE uses a frequency-selective SINC-shaped RF pulse to excite proton spins in a spherical region with the same diameter as the FOV centered at the iso-center. In this case, the spherical excitation region may not contain the apical region due to the patient's varying lung sizes given a fixed FOV, and lower SNR may be acquired in that region. This problem can be reduced by adjusting the FOV according to the patient's lung size. Despite some shortcomings, UTE imaging such as CODE may be a good option for serial lung parenchyma evaluation given the advantages of nonionizing imaging [29]. It is important to establish criteria for evaluating disease severity with MRI alone and to evaluate improvement and progression, particularly for chronic lung disease requiring repeated longitudinal examination.

This study had some limitations. First, with the except of pulmonary nodules, the number of other lesions was very small, limiting the evaluation. In the case of pulmonary nodules, the number of part-solid nodules and ground glass nodules was smaller than that of solid nodules, so their evaluation was also limited. In clinical practice, pulmonary MRI is frequently used to evaluate lung cancer and mediastinal lesions, but is not often used to image other lesions. Despite the small number of cases, our study demonstrates the potential usefulness of pulmonary MRI using the CODE sequence to diagnose thoracic diseases, but further validation is needed in large prospective studies. Second, differences in slice thickness and respiratory gating between VIBE (3.28 mm; breath hold) and CODE (0.82 mm; free breathing) may affect their image quality evaluation, as a thicker slice may result in greater measurement variability and the margin of nodule may be blurred on the thicker slice due to the partial volume effect [26]. Third, because CODE uses a frequency-selective SINC-shaped RF pulse with the excitation bandwidth equal to the readout bandwidth, signals outside the FOV can introduce streak artifacts if the FOV is not large enough (e.g., less than coil sensitivity). A recently proposed volume-selective 3D UTE sequence (VS-UTE) may help to minimize these issues in future studies

[30]. Lastly, compared to VIBE (12.38 sec), CODE (10 min 40 sec) takes more scan time. However, since CODE allows for retrospective respiratory gating, breath-holding is not required, reducing patient burden. In addition, the data acquired in the free-breathing state can reconstructed image not only the expiratory phase but also various respiration phases, so functional image information such as ventilation images can be additionally obtained with a single scan (S1 Data).

In conclusion, our study showed the potential of CODE and VIBE sequences in the evaluation of localized thoracic abnormalities including solid pulmonary nodules. Additionally, because this study well presents the advantages and disadvantages of VIBE as well as CODE in their findings for each anatomy and lung lesion, it is expected to help better interpretation in clinical practice in terms of complementary utilization of multiple sequences, another advantage of lung MRI.

## Supporting information

**S1 File. MRI acquisition and retrospective respiratory gating.**
(PDF)

**S1 Table. Scanner parameters for VIBE and CODE sequences.**
(PDF)

**S2 Table. Qualitative grading system for assessment of images obtained using VIBE and CODE sequences.**
(PDF)

**S1 Data.**
(XLSX)

## Author Contributions

**Conceptualization:** Ho Yun Lee, Jang-Yeon Park.

**Data curation:** So Hyeon Bak, Jinil Park, Ho Yun Lee.

**Formal analysis:** So Hyeon Bak, Jinil Park, Seokwon Lee, Jong Hee Kim, Ho Yun Lee, Jang-Yeon Park.

**Funding acquisition:** Jinil Park, Ho Yun Lee, Jang-Yeon Park.

**Methodology:** So Hyeon Bak, Jinil Park, Seokwon Lee, Ho Yun Lee, Jang-Yeon Park.

**Writing – original draft:** So Hyeon Bak, Jinil Park.

**Writing – review & editing:** Seokwon Lee, Jong Hee Kim, Ho Yun Lee, Jang-Yeon Park.

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
