## [Decision Letter · Decision Letter 0]

31 Oct 2023

PONE-D-23-25050Clinical usability of 3D gradient-echo-based ultrashort echo time imaging: is it enough to facilitate diagnostic decision in real-world practice?PLOS ONE

Dear Dr. Lee,

Thank you for submitting your manuscript to PLOS ONE. After careful consideration, we feel that it has merit but does not fully meet PLOS ONE’s publication criteria as it currently stands. Therefore, we invite you to submit a revised version of the manuscript that addresses the points raised during the review process.

We look forward to receiving your revised manuscript.

Kind regards,

Lorenzo Faggioni, M.D., Ph.D.

Academic Editor

PLOS ONE

“This work was supported by NRF-2020R1A2B5B02002676, NRF-2021R1A4A5032806, NRF-2021R1C1C2008365, and NRF-2022R1A2C1003999.

None of the authors have potential conflicts of financial relationships to disclose.”

Reviewers' comments:

Reviewer's Responses to Questions

**Comments to the Author**

1. Is the manuscript technically sound, and do the data support the conclusions?

Reviewer #1: No

Reviewer #2: Yes

2. Has the statistical analysis been performed appropriately and rigorously? 

Reviewer #1: Yes

Reviewer #2: Yes

3. Have the authors made all data underlying the findings in their manuscript fully available?

Reviewer #1: No

Reviewer #2: Yes

4. Is the manuscript presented in an intelligible fashion and written in standard English?

Reviewer #1: No

Reviewer #2: Yes

5. Review Comments to the Author

Reviewer #1: The paper is interesting in the aim, because would like to assess the utility of CODE sequence in assessment lung abnormalities, but the results are not clear, in fact the differences in qualitative assessment of normal pulmonary structures and specific lesions between CODE and VIBE aren't significant except in signal. You should change the aim in comparison between CT and VIBE/CODE.

- Not clear the sentence: "Thickening of the bronchovascular bundle was considered to be present when the normally tapering bronchovascular bundle leading to the nodule showed distinct and irregular widening"

- the abstract not explain the results and methods

- review figure captions

Reviewer #2: Dear Editor, Dear authors

Thank you for the opportunity to review the article „Clinical usability of 3D gradient-echo-based ultrashort echo time imaging: is it enough to facilitate diagnostic decision in real-world practice?“ By Bak et al.

In the present study, Bak et al. investigate the strengths and weaknesses of lung MRI with regards to UTE sequence techniques in comparison with a conventional T1 VIBE sequence and CT as reference. The authors analyze the image quality using a subjective rating with a 5 point likert score and evaluate quantitative parameters as SNR and CNR. The authors conclude, that the CODE sequence achieves likely similar image quality as compared to VIBE, however, both sequences still have minor image quality compared to the reference. In conclusion, the authors suppose, that CODEhas the potential to be implemented in clinical routine for imaging of lung disease.

The study is well prepared and well written. The data and results appear essentially correct and are presented in a clear fashion, however, I recommend a few changes regarding the interpretation and presentation of data:

1. In the introduction the CODE UTE Sequence is introduced as a „recent development“. CODE has been published in 2012, so I recommend to rephrase this a bit, especially, since not many studies used CODE in the last years.

2. Please emphasize in the Abstract, Introduction and Methods, that CODE is a free breathing acquisition scheme with retrospective gating.

3. Please add in abstract and introduction, that you use the T1 weighted VIBE, to avoid confusion with the prototypical 3D UTE Spiral VIBE that is recently used in many studies regarding Lung MRI.

4. The discussion and especially the conclusion (in main body as well as in the abstract) must emphasize, that the strengths of UTE and T1 VIBE is the depiction of solid (plus) pathologies. Many studies have reported, that MRI lacks the sensitivity to depict emphysema, and furthermore it lacks ability to depict ground glass opacities. It is important, to add this into the conclusions. The statement, that UTE can be used for assessment of lung disease is therefore a bit too unspecific and ultimately maybe even wrong.

6. PLOS authors have the option to publish the peer review history of their article (what does this mean?). If published, this will include your full peer review and any attached files.

Reviewer #1: No

Reviewer #2: No

---

## [Author Response · Author response to Decision Letter 0]

1 Dec 2023

Reviewer #1:

1. The paper is interesting in the aim, because would like to assess the utility of CODE sequence in assessment lung abnormalities, but the results are not clear, in fact the differences in qualitative assessment of normal pulmonary structures and specific lesions between CODE and VIBE aren't significant except in signal. You should change the aim in comparison between CT and VIBE/CODE.

Reviewer 1-1. Thank you for your comment. As mentioned, we have changed the purpose.

Abstract P3. The purpose of our study was to evaluate the current status of the concurrent dephasing and excitation (CODE) ultrashort echo-time sequence and the T1-weighted volumetric interpolated breath-hold examination (VIBE) sequence in the evaluation of thoracic disease by comparing it with the gold standard computed tomography (CT).

2. Not clear the sentence: "Thickening of the bronchovascular bundle was considered to be present when the normally tapering bronchovascular bundle leading to the nodule showed distinct and irregular widening"

Reviewer 1-2. Thank you for your comment. As mentioned, we have modified the sentence. 

Methods P10. Thickening of the bronchovascular bundle was defined as irregular thickening of peribronchovascular interstitium adjacent to the nodule [Aoki T et al. Radiology 2001 & Sung P et al. Clin Radiol 2021].

3. The abstract not explain the results and methods

 Reviewer 1-3. Thank you for your comment. We have modified abstract.

Abstract P3-4. 

Background: With recent advances in magnetic resonance imaging (MRI) technology, the practical role of lung MRI is expanding despite the inherent challenges of the thorax. The purpose of our study was to evaluate the current status of the concurrent dephasing and excitation (CODE) ultrashort echo-time sequence and the T1-weighted volumetric interpolated breath-hold examination (VIBE) sequence in the evaluation of thoracic disease by comparing it with the gold standard computed tomography (CT).

Methods: Twenty-four patients with lung cancer and mediastinal masses underwent both CT and MRI including T1-weighted VIBE and CODE. For CODE images, data were acquired in free breathing and end-expiratory images were reconstructed using retrospective respiratory gating. All images were evaluated through qualitative and quantitative approaches regarding various anatomical structures and lesions (nodule, mediastinal mass, emphysema, reticulation, honeycombing, bronchiectasis, pleural plaque and lymphadenopathy) inside the thorax in terms of diagnostic performance in making specific decisions. 

Results: Depiction of the lung parenchyma, mediastinal and pleural lesion was not significant different among the three modalities (p > 0.05). Intra-tumoral and peritumoral features of lung nodules were not significant different in the CT, VIBE or CODE images (p > 0.05). However, VIBE and CODE had significantly lower image quality and poorer depiction of airway, great vessels, and emphysema compared to CT (p < 0.05). Image quality of central airways and depiction of bronchi/bronchiectasis were significantly better in CODE than in VIBE (p < 0.001 and p = 0.005). In contrast, the depiction of the vasculature was better for VIBE than CODE images (p = 0.003). The signal-to-noise ratio (SNR) and contrast-to-noise ratio (CNR) were significant greater in VIBE than CODE except for SNRlung and SNRnodule (p < 0.05). 

Conclusions: Our study showed the potential of CODE and VIBE sequences in the evaluation of localized thoracic abnormalities including solid pulmonary nodules.

4. Review figure captions

 Reviewer 2-1. Thank you for your comment. We have modified figure caption (figure 2 to 4).

P8. 

Fig 2. Schematic diagram of the VIBE and CODE sequences. (A) The VIBE sequence is basically same as a conventional 3D gradient-echo-imaging sequence, except for asymmetric sampling of the phase-encoding steps with zero padding in the slab-selective direction. (B) The CODE sequence is a 3D radial asymmetric gradient echo-based UTE sequence. Due to the absence of a phase-encoding step, CODE has a much shorter TE and is less sensitive to motion than VIBE.

P15. 

Fig 3. Chest images from a 63-year-old female with lung cancer in the right lower lobe showing spiculated margins, internal air and pleural tags obtained using CT (A), or MRI with VIBE (B) or CODE (C). In a 24-year-old female, CT (D), VIBE (E), and CODE (F) images each show a huge heterogeneously enhanced mass narrowing the lumen of the trachea in the anterior mediastinum. Images from a 71-year-old male indicate pleural plaque (white arrow) in the right hemithorax on CT (G), while both VIBE (H) and CODE (I) imaging also show pleural plaque (white arrow). CODE produced a stronger signal and the plaque is clearly visible.

Fig 4. (A-C) Images from a 75-year-old male with emphysema. Chest CT (A) shows centrilobular and paraseptal emphysema. Although emphysema is visible in images obtained using MRI with VIBE (B) or CODE (C) sequences, the extent of emphysema is underestimated on CODE imaging. (D-F) Images from a 75-year-old male with idiopathic pulmonary fibrosis and lung cancer. Chest CT (D) shows a subpleural nodule in right lower lobe and honeycombing, traction bronchiectasis, and reticulation in the both lower lobe. The VIBE (E) and CODE (F) images also indicate a subpleural nodule and honeycombing in the right lower lobe, but bronchiectasis and reticulation in the left lower lobe were less clear in the VIBE and CODE images as compared with the CT image.

Reviewer #2:

1. In the introduction the CODE UTE Sequence is introduced as a „recent development“. CODE has been published in 2012, so I recommend to rephrase this a bit, especially, since not many studies used CODE in the last years.

Reviewer 2-1. Thank you for your comment. We agree that the word ‘recent development’ is not appropriate because CODE was developed in 2012. Accordingly, we have eliminated “recently proposed” in the revised manuscript. In fact, there have not been many studies using CODE since 2012, but two previous studies (References 11 and 12) were already mentioned in the next paragraph of the original manuscript. So, we changed the sentence, just removing the phrase “recently proposed.” We would appreciate your understanding in this regard.

Introduction P5. On the other hand, asymmetric gradient echoes are acquired in a 3D UTE sequence called concurrent dephasing and excitation (CODE)

2. Please emphasize in the Abstract, Introduction and Methods, that CODE is a free breathing acquisition scheme with retrospective gating.

Reviewer 2-2. Thank you for your comment. As suggested, we have updated the Abstract, Introduction, and Methods by emphasizing that CODE is a free breathing acquisition scheme with retrospective gating.

Abstract P3. For CODE images, data were acquired in free breathing and end-expiratory images were reconstructed using retrospective respiratory gating.

Introduction P6. CODE images are obtained in free breathing with retrospective respiratory gating, alleviating the burden on patients with lung disease, especially those who have difficulty holding their breath.

Methods (MRI acquisition) P8. CODE data were acquired under free breathing conditions. The acquired data included self-navigation echoes, from which respiratory motion signals were extracted for retrospective respiratory gating.

Methods (image reconstruction) P8-9. End-expiratory CODE images were reconstructed via retrospective respiratory gating, i.e. by selecting the radial views corresponding to the end-expiratory phase of the respiratory signal. End- expiratory images were chosen because expiration typically lasts longer than inspiration and therefore end-expiratory data has a greater number of radial views for image reconstruction.

3. Please add in abstract and introduction, that you use the T1 weighted VIBE, to avoid confusion with the prototypical 3D UTE Spiral VIBE that is recently used in many studies regarding Lung MRI.

Reviewer 2-3. Thank you for your comment. As suggested, we have updated the Abstract and Introduction.

Abstract P3

 Background: The purpose of our study was to evaluate the current status of the concurrent dephasing and excitation (CODE) ultrashort echo-time sequence and the T1-weighted volumetric interpolated breath-hold examination (VIBE) sequence in the evaluation of thoracic disease by comparing it with the gold standard computed tomography (CT).

 Methods: Twenty-four patients with lung cancer and mediastinal masses underwent both CT and MRI including T1-weighted VIBE and CODE.

Introduction P6. The purpose of this study was to compare CODE and T1-weighted VIBE, the most widely used post-contrast MR sequence, with the gold-standard CT to determine the applicability of MR in the evaluation of thoracic disease and ultimately to confirm the feasibility of CODE in real-world practice.

4. The discussion and especially the conclusion (in main body as well as in the abstract) must emphasize, that the strengths of UTE and T1 VIBE is the depiction of solid (plus) pathologies. Many studies have reported, that MRI lacks the sensitivity to depict emphysema, and furthermore it lacks ability to depict ground glass opacities. It is important, to add this into the conclusions. The statement, that UTE can be used for assessment of lung disease is therefore a bit too unspecific and ultimately maybe even wrong.

Reviewer 2-4. Thank you for your comment. As mentioned, we have modified abstract and discussion. 

Abstract P4. Conclusions: Our study showed the potential of CODE and VIBE sequences in the evaluation of localized thoracic abnormalities including solid pulmonary nodules.

Discussion P18. The current study shows that the depiction and morphologic evaluation of solid nodule in CODE and VIBE were not significant different from that in CT image. The depiction of part-solid nodules and purely ground-glass opacity nodules may be underestimated by MR [Bak SH et al. Precis Future Med 2022 & Wielputx MO et al AJR 2018], yet the number of part-solid nodules and purely ground-glass opacity nodules was too small to evaluate nodule detection and morphologic characterization in our study.

Discussion P21. In conclusion, our study showed the potential of CODE and VIBE sequences in the evaluation of localized thoracic abnormalities including solid pulmonary nodules. Additionally, because this study well presents the advantages and disadvantages of VIBE as well as CODE in their findings for each anatomy and lung lesion, it is expected to help better interpretation in clinical practice in terms of complementary utilization of multiple sequences, another advantage of lung MRI.

---

## [Decision Letter · Decision Letter 1]

18 Dec 2023

Clinical usability of 3D gradient-echo-based ultrashort echo time imaging: is it enough to facilitate diagnostic decision in real-world practice?

PONE-D-23-25050R1

Dear Dr. Lee,

We’re pleased to inform you that your manuscript has been judged scientifically suitable for publication and will be formally accepted for publication once it meets all outstanding technical requirements.

Kind regards,

Lorenzo Faggioni, M.D., Ph.D.

Academic Editor

PLOS ONE

**Comments to the Author**

1. If the authors have adequately addressed your comments raised in a previous round of review and you feel that this manuscript is now acceptable for publication, you may indicate that here to bypass the “Comments to the Author” section, enter your conflict of interest statement in the “Confidential to Editor” section, and submit your "Accept" recommendation.

Reviewer #2: All comments have been addressed

2. Is the manuscript technically sound, and do the data support the conclusions?

Reviewer #2: Yes

3. Has the statistical analysis been performed appropriately and rigorously? 

Reviewer #2: Yes

4. Have the authors made all data underlying the findings in their manuscript fully available?

Reviewer #2: Yes

5. Is the manuscript presented in an intelligible fashion and written in standard English?

Reviewer #2: Yes

6. Review Comments to the Author

Reviewer #2: Thank you, all my recommendations have been addressed. I recommend a final proof read. In the abstract "significant" is used as an adjective instead of an adverb "significantly" in multiple instances, this must be changed before publication.

7. PLOS authors have the option to publish the peer review history of their article (what does this mean?). If published, this will include your full peer review and any attached files.

Reviewer #2: No

---

## [Editor Report · Acceptance letter]

29 Apr 2024

PONE-D-23-25050R1 

PLOS ONE

Dear Dr. Lee, 

I'm pleased to inform you that your manuscript has been deemed suitable for publication in PLOS ONE. Congratulations! Your manuscript is now being handed over to our production team.

Kind regards, 

on behalf of

Dr. Lorenzo Faggioni 

Academic Editor

PLOS ONE